# Bioassay-Guided Isolation of Two Eudesmane Sesquiterpenes from *Lindera strychnifolia* Using Centrifugal Partition Chromatography

**DOI:** 10.3390/molecules26175269

**Published:** 2021-08-30

**Authors:** Ji Hoon Kim, Je-Seung Jeon, Jung Hoon Kim, Eun Ju Jung, Yun Jung Lee, En Mei Gao, Ahmed Shah Syed, Rak Ho Son, Chul Young Kim

**Affiliations:** 1College of Pharmacy, Institute of Pharmaceutical Science and Technology, Hanyang University, Ansan 15588, Korea; gg890718@gmail.com (J.H.K.); jsjeoncy@gmial.com (J.-S.J.); evveje@naver.com (J.H.K.); jejs2@naver.com (E.J.J.); sopihya@naver.com (Y.J.L.); rhdmsal@hanyang.ac.kr (E.M.G.); sonnaco@huons.com (R.H.S.); 2Department of Pharmacognosy, Faculty of Pharmacy, University of Sindh, Jamshoro 76080, Pakistan; shahahmed454@gmail.com; 3R&D Center, Huons Co., Ltd., Ansan 15588, Korea

**Keywords:** *Lindera strychnifolia*, centrifugal partition chromatography, linderolide V, eudesmane sesquiterpenes

## Abstract

In this study, a centrifugal partition chromatography (CPC) separation was applied to identify antioxidant-responsive element (ARE) induction molecules from the crude extract of *Lindera strychnifolia* roots. CPC was operated with a two-phase solvent system composed of *n*-hexane-methanol-water (10:8.5:1.5, *v*/*v*/*v*) in dual mode (descending to ascending), which provided a high recovery rate (>95.5%) with high resolution. Then, ARE induction activity of obtained CPC fractions was examined in ARE-transfected HepG2 cells according to the weight ratios of the obtained fractions. The fraction exhibiting ARE-inducing activity was further purified by preparative HPLC that led to isolation of two eudesmane type sesquiterpenes as active compounds. The chemical structures were elucidated as linderolide U (**1**) and a new sesquiterpene named as linderolide V (**2**) by spectroscopic data. Further bioactivity test demonstrated that compounds **1** and **2** enhanced ARE activity by 22.4-fold and 7.6-fold, respectively, at 100 μM concentration while 5 μM of sulforaphane induced ARE activity 24.8-fold compared to the control.

## 1. Introduction

Natural products are a pool of molecules that attract pharmaceutical interest. To search interesting bioactive compounds from natural extracts, performing a bioassay-guided fractionation that gears toward silica-based chromatographic steps is common. During the isolation workflow, however, innate physicochemical properties of its solid-like packing materials often bring about loss of activity or failure in isolation concomitant with insufficient concentration of target compounds for bioactivity test [1,2]. Thus, a fractionation system that enhances separation efficiency and total sample recovery while lowering the risk in sample denaturation is needed. For this purpose, counter-current separations are an efficient chromatographic tool to combine with bioactivity tests. Centrifugal partition chromatography (CPC) is a type of support-free liquid–liquid chromatography technique that uses an immiscible two-phase solvent system to compose stationary and mobile phase. CPC offers many advantages such as a higher sample loading capacity, the absence of sample loss due to irreversible sample absorption to the solid column, a flexible operation mode, and the choice of a wide range of solvent systems [3,4,5,6,7,8,9,10,11,12], so it can achieve the fractionations with sufficient amount to further in vitro bioactivity tests [11,12].

*Lindera strychnifolia* Fernandez-Villar (Lauraceae) is an evergreen shrub that is widely distributed in Asia. As *Lindera* root is known to stimulate Qi circulation, eliminate cold and pain, it has been used in the treatment of stomach and renal diseases, neuralgia, and rheumatism in traditional Oriental medicine [13]. Recent studies also reported antioxidant, anti-diabetic, and anti-inflammatory effects of *Lindera* root. Sesquiterpenes are major constituents of this plant, and alkaloids and tannins are also isolated [14,15,16,17]. The crude extract of the root of *L. strychnifolia* enhanced the antioxidant response element (ARE) activity in an ongoing screening for natural products in HepG2-ARE luciferase assay.

In this study, the dual-mode CPC was applied to obtain high-resolution fractions of *L. strychnifolia* extract and obtained fractions were tested in ARE-inducing activity in accordance with the weight ratio of fractions. The active fraction was further purified by preparative HPLC to isolate two eudesmane sesquiterpenes exhibiting ARE-inducing activities.

## 2. Results

### 2.1. Isolation of Active Compounds Using Dual-Mode CPC and HPLC

When the preliminary experiment was carried out using linear-gradient CPC to identify active peaks from nonpolar to polar fractions, nonpolar fraction-exerted ARE induction activity and comparison of an active fraction and *n*-hexane extract was revealed similar HPLC chromatograms (Appendix A). Therefore, *n*-hexane extract was chosen for further purification. HPLC analysis in Figure 1 presented several major peaks **a**–**e**, which were used for the calculation of partition coefficient (*K* value).

After calculation of *K* values of major peaks **a**–**e**, a two-phase solvent system with *n*-hexane-methanol-water (10:8.5:1.5, *v*/*v*/*v*) was selected (Table 1).

Dual-mode CPC was applied to recover all introduced samples with high-resolution. The upper organic phase was firstly eluted as the mobile phase. After 230 min operation in ascending mode, elution was changed to descending mode to recover remaining samples in the rotor and maintained to 345 min.

As shown in Figure 2, ten fractions (**H1**–**H10**) were obtained: **H1** (30–45 min, 940.8 mg), **H2** (45–100 min, 964.1 mg), **H3** (100–170 min, 89.1 mg), **H4** (170–230 min, 114.0 mg), **H5** (230–235 min, 550.4 mg), **H6** (235–250 min, 417.5 mg), **H7** (250–255 min, 45.2 mg), **H8** (255–270 min, 339.8 mg), **H9** (270–305 min, 1226.5 mg), and **H10** (305–345 min, 88.9 mg). The total sum of each fraction was 4776.3 mg out of 5.0 g of crude sample, exhibiting 95.5% recovery rate. Next, each fraction was evaluated for the ARE-inducing activity at a concentration of 30 μg/mL, and ten fractions were evaluated at the concentration applied at each assigned weight ratio (Table 2).

As shown in Figure 3, the results showed that fraction **H9** exerted the highest activity. HPLC analysis revealed that **H9** fraction contained two major peaks, **1** and **2**. Further purification by prep-HPLC led to the isolation of compounds **1** and **2**. The following section will describe structural elucidations.

### 2.2. Structural Elucidation of Compounds ***1*** and ***2***

Chemical structures of purified compounds were determined by spectroscopic methods, including NMR (^1^H, ^13^C NMR, COSY, HSQC, HMMC, NOESY and Mosher’s method) and HR-ESI-MS spectral data (Table 3).

Compound **1** was isolated as a brownish powder. Its molecular formula was determined as C_16_H_20_O_4_ by the HR-ESI-MS ion at *m/z* 299.1242 [M + Na]^+^. Analysis of the ^13^C and HSQC spectra of **1** revealed 16 carbon signals that attributed to two methyls (at *δ*_C_ 17.9 and 20.2), one methoxy (at *δ*_C_ 52.8), two methylenes (at *δ*_C_ 17.6, 54.4), one *exo*-methylene (at *δ*_C_ 108.6), three methines (at *δ*_C_ 29.1, 24.4 and 67.2), one oxygenated methine (at *δ*_C_ 65.9), and six quaternary carbons. In the ^1^H NMR spectrum, two methyl signals at *δ*_H_ 0.74 and 2.10 (each 3H), one methoxyl at *δ*_H_ 3.72 (3H, s), one hydroxymethine at *δ*_H_ 4.42 (1H, dd, *J* = 11.0, 0.9), and three methines at *δ*_H_ 1.45 (1H, ddd, *J* = 8.0, 7.1, 3.6), 2.04 (1H, m) and 2.79 (1H, ddd, *J* = 11.0, 3.2. 2.2) were observed. Besides, the presence of *exo*-methylene was easily deduced from resonances at *δ*_H_ 5.10 (1H, dt, *J* = 2.6, 1.4 Hz) and 5.24 (1H, d, *J* = 1.4 Hz) connected to *δ*_C_ 108.6 in the HSQC spectrum. In the ^1^H-^1^H COSY spectrum, the connections of -CH(H-1)-CH_2_(H-2)-CH(H-3)- and CH(H-5)-CH(H-6) were observed; especially, long-range couplings of H-15 and H-5, H-13 and H-6 were detected. In the HSQC spectrum, one oxymethine carbon was observed at *δ*_C_ 65.9, connected to *δ*_H_ 4.42 (1H, dd, *J* = 11.0, 0.9). Besides, three methines were also revealed at *δ*_C_ 29.1, 24.4 and 67.2, corresponding to *δ*_H_ 1.45 (1H, dd, *J* = 11.0, 1.2), 2.04 and 2.79, respectively. In the HMBC spectrum, correlations were observed between from H-15 to C-3/C-3/C-4, from H-14 to C-1/C-5/C-9/C-10, from H-13 to C-7/C-11/C-12, from H-2 to C-1/C-3/C-10 and from H-9 to C-8/C-10. The HMBC spectrum showed a correlation between *δ*_H_ 2.10 (H-13), 3.72 (OCH_3_) and a carbonyl resonance at *δ*_C_ 173.5 (C-12), and between *δ*_H_ 2.58 (H-9), 4.42 (H-6) and a carbonyl signal at *δ*_C_ 202.36 (C-8), showing similarity to those of a side chain in linderolides L and M [15]. The relative stereochemistry of 1 was determined by NOESY, which showed correlations between Me-14 and H-6, which suggested that these protons exist on the same side. The NOESY correlation between H-6 and CH_3_-13 suggested a *Z*-configuration of the 7,11 double bond. The absolute configuration at the 6-position of **1** was determined to be *S* by a modified Mosher’s method (Appendix A) [18]. Based on the obtained data, compound 1 was determined as linderolide U (Figure 4). Recently, linderolide U was identified in this plant [19].

Linderolide V (2) was isolated as a brown powder, and its molecular formula was determined to be C_15_H_18_O_3_ based on HR-QTOF/MS at *m/z* 269.1231 [M + Na]^+^. The ^13^C and HSQC spectra revealed 15 carbon signals that attributed to two methyls (at *δ*_C_ 17.3 and 14.3), three methylenes (at *δ*_C_ 18.56, 38.1, 108.6), three methines (at *δ*_C_ 28.4, 23.7 and 67.4), one oxymethine (at *δ*_C_ 64.1), and five quaternary carbons. In the ^1^H NMR spectrum, two methyl signals at *δ*_H_ 0.71 (s) and 1.33 (d), one hydroxymethine at *δ*_H_ 4.10 (1H, d, *J* = 9.6), three methines at *δ*_H_ 1.52 (1H, dt, *J* = 7.5, 3.9), 1.98 (1H, t, 9.4) and 3.37 (1H, m), and characteristic *exo*-methylene resonances at *δ*_H_ 5.06 and 5.20 were observed. The correlations of –CH(H-1)–CH_2_(H-2)–CH(H-3)- and –CH(H-5)–CH(H-6) were observed in the ^1^H-^1^H COSY spectrum.

In the HSQC spectrum, one oxymethine carbon was observed at *δ*_C_ 64.1, connected to *δ*_H_ 4.10 (1H, d, *J* = 9.62). Besides, three methines were also revealed at *δ*_C_ 28.4, 23.7 and 67.4, corresponding to *δ*_H_ 1.52 (1H, dt, *J* = 7.5. 3.9), 1.98 and 2.69, respectively. In the HMBC spectrum, correlations were observed from H-15 to C-3/C-5/C-4, from H-14 to C-1/C-5/C-10, from H-13 to C-7/C-11/C-12, from H-2 to C-1/C-3/C-4/C-10 and from H-9 to C-2/C-5/C-7/C-8/C-10, from H-6 to C-7/C-8. The chemical structure of **2** was similar to the previously reported compound, shizukanolide [20], but the double bond position C-7/C11 was changed to C7/C8 (Figure 4). Based on the obtained data, compound **2** was determined as shown in Figure 4 and was named linderolide V.

The relative stereochemistry of **2** was determined by NOESY, which showed correlations between CH_3_-14 and H-6 which suggested that these protons exist on the same side. CH_3_-13 and H-5 also showed a correlation that means these protons are on the same side. However, CH_3_-13 and CH_3_-14 did not show any correlations that are located on the opposite side (Figure 5). The coupling constant between H-5/H-6 was 9.6 Hz which supported that H-5 and H-6 were located on the other side. In addition, the Δ*δ* values between the (*S*) and (*R*)-MTPA esters indicated the 6*S* configuration for **2** (Appendix A) [18].

### 2.3. Induction of Antioxidant Response Element by Compounds ***1*** and ***2***

The ARE-inducing ability of the isolated compounds **1** and **2** from an active fraction (**H9**) were assessed by luciferase assay in HepG2 cells at a serial concentration of 3, 10, 30, and 100 μM. Linderolide U (1) and linderolide V (2) enhanced ARE activity in a dose-dependent manner (Figure 6). At 100 μM concentration, Compounds **1** and **2** enhanced ARE activity 22.4-fold and 7.6-fold, respectively, while 5 μM of sulforaphane induced ARE activity 24.8-fold. These results indicated that compounds **1** and **2** were active principles for ARE induction of *L. strychnifolia*.

## 3. Materials and Methods

### 3.1. Apparatus

An Armen fully integrated SCPC-100 + 1000 CPC spot instrument (Armen Instruments, St-Ave, France) was used in this study. This instrument is a fully automated system consisting of a CPC column compartment (1000 mL rotor made of 21 stacked disks with a total of 1512 twin cells), a pump, an injector, a UV/vis detector, a fraction collector, a digital screen flat PC and Armen Glider CPC software. The HPLC analysis was performed by an Agilent 1260 HPLC system (Agilent Technologies, Palo Alto, CA, USA): G1312C binary pump, a G1329B autosampler, a G1315D DAD detector, a G1316A column oven, and ChemStation software.

### 3.2. Chemicals and Reagents

All solvents used for the CPC were of analytical grade and purchased from Daejung Chemical (Korea). The HPLC-grade solvents were obtained from Fisher Scientific (Pittsburgh, PA, USA). The *L. strychnifolia* roots, well dried and sliced around 2 mm, were purchased from the Kyungdong Oriental herbal market, Seoul, Republic of Korea, in January 2018. A voucher specimen (HYUP-LS-001) was deposited in the Herbarium of the College of Pharmacy, Hanyang University (Appendix A).

### 3.3. Preparation of Crude Samples

Dried and coarsely ground sample (600 g of *L. strychnifolia* roots) was extracted with methanol by reflux for 2 h. The extraction was repeated three times and combined. The filtrate was evaporated under reduced pressure using a rotary evaporator to obtain 12.5 g of crude extract. Ten grams of crude extract was dissolved in water and extracted with 500 mL *n*-hexane three times. After dryness, 5.4 g of *n*-hexane extract was obtained.

### 3.4. HPLC Analysis

The crude extract and its CPC peak fraction were profiled by HPLC equipped with a Capcellpak UG120 C18 column (4.6 × 250 mm, 5 μm, Shiseido, Tokyo, Japan). The mobile phase was composed of acetonitrile containing 0.1% formic acid (A) and water containing 0.1% formic acid (B). The gradient elution conditions were as follows: 0–10 min, 10–50% A; 10–45 min, 50–95% A; and 50 min, 95% A. The column temperature was 40 °C, the mobile phase flow rate was 1 mL/min, the detection wavelength was 230 and 254 nm and the injection volume was 10 µL.

### 3.5. Selection of Solvent System

Because *n*-hexane extract was nonpolar, a series of solvent systems of *n*-hexane–methanol-water were tested. Briefly, approximately 2 mg of the sample was added to each test tube, 2 mL of each phase of a pre-equilibrated two-phase solvent system was added, and they were thoroughly mixed. After equilibration, 100 µL of the upper and lower phases were added to 900 µL methanol, and 10 µL were analyzed by HPLC at 230 nm. The *K* values of major peaks (**a**–**e**) were described as the peak area of each compound in the upper stationary phase divided by that of the lower mobile phase.

### 3.6. CPC Operation for Active n-Hexane Extract

*n*-Hexane extract was purified by CPC operation with a two-phase solvent system composed of *n*-hexane-methanol-water (10:8.5:1.5, *v*/*v*/*v*) according to the *K* values of major peaks **a**–**e** (Table 1). The lower aqueous phase was used as mobile phase and the upper organic phase as stationary phase with ascending mode. The 1000 mL volume of the CPC rotor was filled with a lower layer at 50 mL/min in ascending mode at a speed of 500 rpm. Then, the rotation speed of the rotor was accelerated to 1000 rpm, and the upper layer as the mobile phase was carried into the rotor in descending mode at a flow rate of 10 mL/min. After the equilibration between the upper phase and lower phase was established, the sample solution (5.0 g) was loaded into CPC system, and the effluents were continuously monitored by a UV detector at 230 and 254 nm. After 230 min operation in ascending mode, elution was changed to descending mode to recover remaining samples in the rotor and maintain to 330 min.

### 3.7. Isolation and Structural Elucidation of Active Compounds ***1*** and ***2***

After CPC operation and evaluation of ARE-inducing activity, subfraction **H9** was further purified by preparative HPLC. HPLC was carried out using a Gilson 321 pump, a Waters 2487 detector, an RS Tech HECTOR C18 column (5 µm, 250 × 21.2 mm, RS Tech Corp, Cheongju, South Korea). Isocratic elution mode (40% acetonitrile) at a flow rate of 10 mL/min was used and monitored at 230 and 254 nm. Compounds **1** (42.6 mg) and **2** (13.8 mg) were obtained at a retention time of 23.8 and 35.5 min, respectively. 1D- (^1^H, 400 MHz, ^13^C, 100 MHz) and 2D-NMR (^1^H-^1^H COSY, HSQC, HMBC, and NOESY) were measured on a Bruker model digital Advance III 400 NMR for structural elucidation. The NMR data were described in Table 3.

### 3.8. Preparation of the (R)- and (S)-MTPA Ester Derivatives

To assign the absolute configuration to the molecules, the (*S*)- and (*R*)-methoxy-α-(trifluoromethyl)phenylacetyl (MTPA) ester derivatives of **1** and **2** were synthesized in deuterated pyridine (pyridine-d_5_) from (*R*)-(+)-MTPA-Cl and (*S*)-(−)-MTPA-Cl, respectively [18]. Compound **1** (2.0 mg) was dissolved in pyridine-d_5_ (6 mL) and divided into two NMR tubes (each 1 mL). (*R*)-MTPA-Cl (4 μL) was added and reacted at room temperature for 3 h to yield (*S*)-MTPA ester derivative (**1S**). Another tube, (*S*)-MTPA-Cl (4 μL), was added and reacted at room temperature to yield (*R*)-MTPA ester derivative (**1R**), and the ^1^H NMR spectrum was recorded on 400 MHz NMR. Similarly (*S*)- and (*R*)-MTPA ester derivatives of **2** were prepared as **2S** and **2R**.

### 3.9. Assay of ARE-Inducing Activities

HepG2-ARE cells were seeded at a density of 1 × 10^5^ cells/well in 24-well plates for 24 h. The cells were starved for 12 h when they grew to around 80% confluency and were exposed to crude extract, CPC fractions and purified compounds for an additional 24 h. After that, the cells were lysed with 120 μL of passive lysis buffer (Promega, Madison, WI, USA) in an ice rack and transferred in the 1.5-mL tube. The tubes were centrifuged at 1000 rpm for 3 min. Each supernatant (30 μL) in the centrifuged tube was reacted with 60 μL of luciferase assay substrate (Promega, Madison, WI, USA) in the white 96 well plate. Finally, luminescence was measured by an EnSpire multimode plate reader (PerkinElmer, Waltham, MA, USA). DMSO (below 0.1%) was used as a vehicle, which was a negative control. Sulforaphane (5 μM) (Calbiochem, Darmstadt, Germany) was used as a positive control. The ARE-luciferase activity was normalized to total protein, determined using BCA protein assay kit with BSA as a standard (Pierce No. 23227).

### 3.10. Statistical Analysis

All data are reported as means ± S.E. The statistical significance of differences between treatments was assessed using the Student’s *t*-test. A probability value less than 0.05 or 0.01 was considered significant.

## 4. Conclusions

In the present study, a bioassay-guided isolation method was developed using CPC. To do this, a comprehensive dual-mode CPC was conducted to fractionate bioactive molecules from *L. strychnifolia* root extract. Its higher recovery rate (>95.5%) allowed unbiased bioactivity tests by avoiding irreversible sample adsorption and denaturation that conventional chromatography methods have. After determining active fraction, the constituents were further purified by preparative HPLC to yield linderolide U (**1**) and a new sesquiterpene, linderolide V (**2**) responsible for ARE-inducing activity of *L. strychnifolia* roots. However, as the identified active compounds exist in trace amounts in *L. strychnifolia*, quantitative analysis methods for these compounds are still required. The significance of this study is that the CPC method is an effective screening tool to unearth known or novel bioactive compounds from natural products.

## Figures and Tables

**Figure 1 molecules-26-05269-f001:**
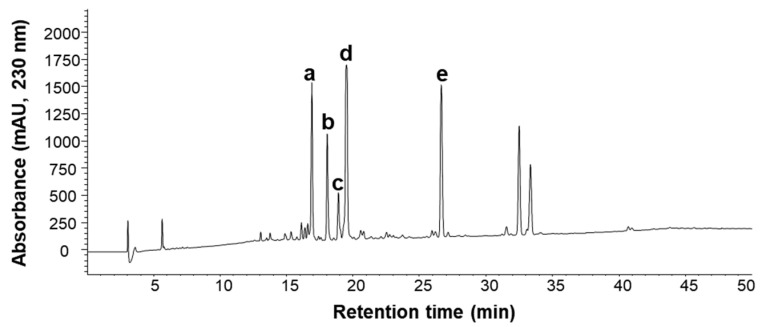
HPLC chromatogram of *n*-hexane extract of *L. strychnifolia*.

**Figure 2 molecules-26-05269-f002:**
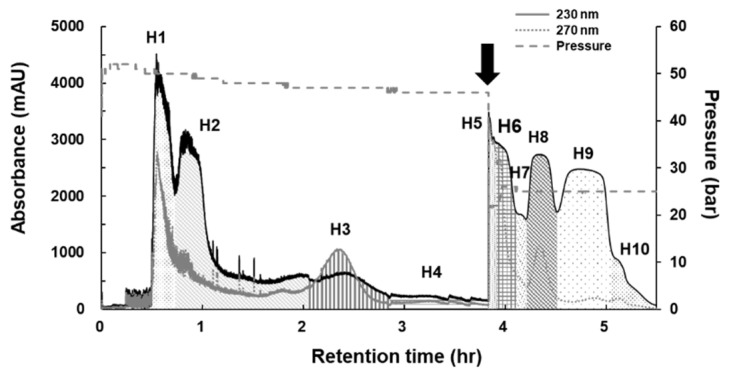
Dual mode CPC chromatogram of *n*-hexane extract of *L. strychnifolia*. The details are described in Section 3.6 CPC procedure.

**Figure 3 molecules-26-05269-f003:**
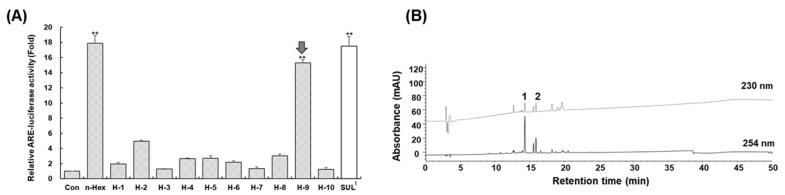
The relative antioxidant response element (ARE)-luciferase activities (**A**) of the CPC-fractions **H1H10** and HPLC chromatogram (**B**) of active fraction **H9**. The ARE induction activities were evaluated in ARE-HepG2 cells at concentrations applied at each assigned weight ratio (based on 30 μg/mL of crude extract). Data are presented as the mean ± S.E. (*n* = 3). ** *p* < 0.01 (compared with the vehicle-treated control). ^1^ Sulforaphane was treated 5 μM as positive control.

**Figure 4 molecules-26-05269-f004:**
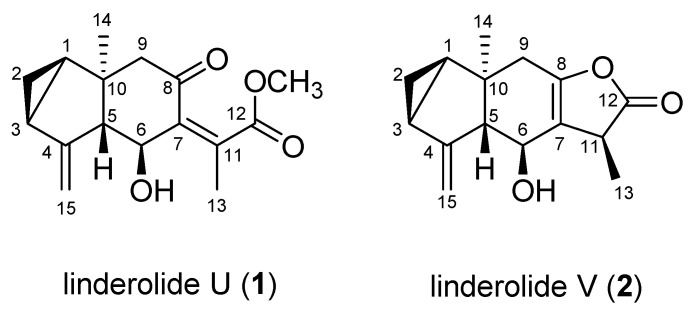
Chemical structures of isolated compounds **1** and **2**.

**Figure 5 molecules-26-05269-f005:**
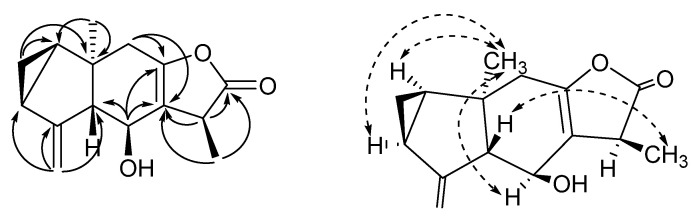
Key HMBC (H
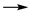
C) and NOESY (H
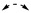
H) correlations of compound **2**.

**Figure 6 molecules-26-05269-f006:**
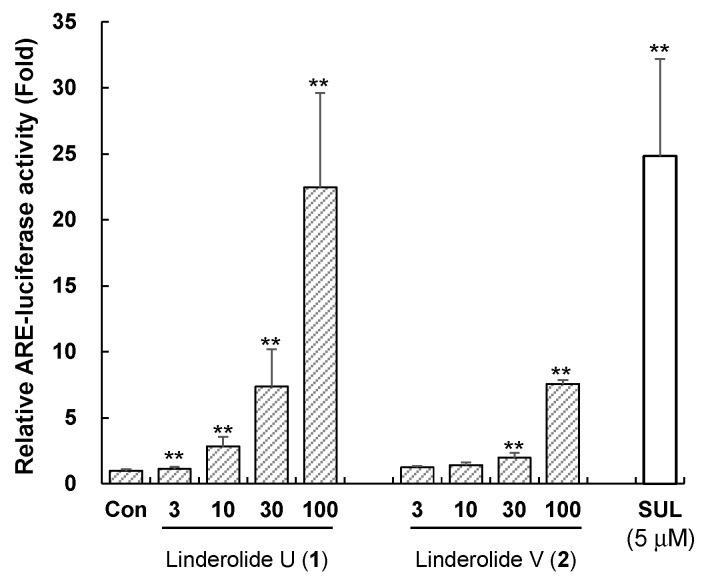
The relative antioxidant response element (ARE)-luciferase activities of isolated compounds **1** and **2**. The ARE induction activities were evaluated in ARE-HepG2 cells at concentrations applied at each assigned weight ratio (based on 30 μg/mL crude extract. Data are presented as the mean ± S.E. (*n* = 3). ** *p* < 0.01 (compared with the vehicle-treated control). SUL: Sulforaphane was treated as a positive control.

**Table 1 molecules-26-05269-t001:** The *K* values of major peaks **a**–**e** in *n*-hexane extract in different solvent systems.

Solvent System (*v*/*v*/*v*) *n*-Hexane:Methanol:Water	Partition Coefficient (*K* ^1^)				
	Peak a	Peak b	Peak c	Peak d	Peak e
10:9.5:0.5	5.47	4.22	3.90	2.86	0.67
10:9:1	5.02	3.51	3.67	2.15	0.45
10:8.5:1.5	4.28	2.65	2.78	1.84	0.29
10:8:2	2.29	1.54	1.46	1.06	0.17

^1^ *K* value = peak area of upper phase/peak area of lower phase.

**Table 2 molecules-26-05269-t002:** Fraction weights and calculated concentrations.

Fractions	H1	H2	H3	H4	H5	H6	H7	H8	H9	H10	Sum
Weight (mg)	940.8	964.1	89.1	114.0	550.4	417.5	45.2	339.8	1226.5	88.9	4776.3
Weight ratio (%)	19.7	20.2	1.9	2.4	11.5	8.7	0.9	7.1	25.7	1.9	100
Treatment (µg/mL)	5.91	6.06	0.56	0.72	3.46	2.62	0.28	2.13	7.70	1.86	30

**Table 3 molecules-26-05269-t003:** ^1^H (400 MHz) and ^13^C NMR (100 MHz) data for compounds **1** and **2** in CD3OD (δ in ppm, *J* values in parentheses).

No.	Linderolide U (1)			Linderolide V (2)		
	^1^H (*J* in Hz)	^13^C		^1^H (*J* in Hz)	^13^C	
1	1.45 (ddd, *J* = 8.0, 7.1, 3.6)	29.1	CH	1.52 (dt, *J* = 7.5, 3.9)	28.4	CH
2	0.90 (ddd, *J* = 8.9, 8.0, 5.2), 0.81 (dt, *J* = 5.2, 3.6)	17.6	CH_2_	0.88 (ddd, *J* = 10.7, 7.9, 4.2)	18.5	CH_2_
3	2.04 (m)	24.4	CH	1.98 (t, *J* = 9.4)	23.7	CH
4		152.0	C		151.3	C
5	2.79 (ddd, *J* = 11.0, 3.2, 2.2)	67.2	CH	2.69 (dt, *J* = 9.6, 2.6)	67.4	CH
6	4.42 (dd, *J* = 11.0, 0.9)	65.9	CH	4.10 (d, *J* = 9.6)	64.1	CH
7		139.9	C		120.9	C
8		202.3	C		151.2	C
9	2.58 (dd, *J* = 4.3, 0.9)	54.4	CH_2_	2.59 (dt, *J* = 16.3, 3.5), 2.36 (d, *J* = 16.3)	38.1	CH_2_
10		37.7	C		40.6	C
11		146.0	C	3.37 (m)	39.8	CH
12		173.5	C		181.7	C
13	2.10 (d, *J* = 0.9)	17.9	CH_3_	1.33 (d, *J* = 7.6)	14.3	CH_3_
14	0.74 (s)	20.2	CH_3_	0.71 (s)	17.3	CH_3_
15	5.24 (d, *J* = 1.4), 5.10 (dt, *J* = 2.7, 1.4)	108.6	CH_2_	5.20 (brs), 5.06 (d, *J* = 1.2)	108.6	CH_2_
OCH_3_	3.72 (s, 3H)	52.8	CH_3_			

## Data Availability

Not applicable.

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
