# Peer review of "Bioassay-Guided Isolation of Two Eudesmane Sesquiterpenes from Lindera strychnifolia Using Centrifugal Partition Chromatography"

_molecules, 2021, doi:10.3390/molecules26175269_

Round 1

Reviewer 1 Report

Comments

The present study uses a centrifugal partition chromatography system and a bioguided isolation method to isolate and identify sesquiterpenes from the crude extract of Lindera strychnifolia roots.

The topic of the paper is relevant to the Journal, the experiments are conducted correctly and the paper is well written.

Corrections
In table 2, compound 2 is called linderolide U, subsequently in the text compound 2 is called linderolide V. Correct the error.

Readers might be interested in the characteristics of the sample used.
Implement sample information in Materials and Methods, such as:
1) The product is commercial or sample is collected in a specific geographical area.
2) The sample was purchased dry or fresh (in this case briefly describe the drying phases)
3) The sample was purchased ground or whole (in this case briefly describe the grinding phases)

The conclusions are too short and linear, they only report the information already present in the text.
The authors should express the usefulness of this research by making the results obtained more interesting.

Author Response

The present study uses a centrifugal partition chromatography system and a bioguided isolation method to isolate and identify sesquiterpenes from the crude extract of Lindera strychnifolia roots.

The topic of the paper is relevant to the Journal, the experiments are conducted correctly and the paper is well written.

 Corrections
In table 2, compound 2 is called linderolide U, subsequently in the text compound 2 is called linderolide V. Correct the error.

- We corrected.

Readers might be interested in the characteristics of the sample used.
Implement sample information in Materials and Methods, such as:
1) The product is commercial or sample is collected in a specific geographical area.

2) The sample was purchased dry or fresh (in this case briefly describe the drying phases).

3) The sample was purchased ground or whole (in this case briefly describe the grinding phases)

- We described the sample state in the text and marked in blue colors. The samples are obtainable in commercial and well dried and sliced around 2 mm thick. And the picture of samples was added in Supplementary Material in Figure S1.

The conclusions are too short and linear, they only report the information already present in the text.
The authors should express the usefulness of this research by making the results obtained more interesting.

- We add the merits of this experiments: The significance of this study is that CPC method is an effective screening tool to unearth known or novel bioactive compounds from natural products.

Reviewer 2 Report

The authors describe about the isolation and structure determination of novel compounds from the root extract of L. strychnifolia in this article. The authors use CPC to efficiently separate and analyze using antioxidant activity as an index. This article is considered  novel and to be accurate. Therefore, I think this article is suitable for publication.

I think this manuscript can be published as it is. The only improvement is a quantitative analysis of the absolute amount of isolated compounds in the extract, because other active compounds may be present. However, since this point is the next research stage, I think it is better to write it down in the manuscript as the next research subject if possible. Kindly

Author Response

Reviewer 2

The authors describe about the isolation and structure determination of novel compounds from the root extract of L. strychnifolia in this article. The authors use CPC to efficiently separate and analyze using antioxidant activity as an index. This article is considered novel and to be accurate. Therefore, I think this article is suitable for publication.

I think this manuscript can be published as it is. The only improvement is a quantitative analysis of the absolute amount of isolated compounds in the extract, because other active compounds may be present. However, since this point is the next research stage, I think it is better to write it down in the manuscript as the next research subject if possible. Kindly

- We are appreciated reviewer’s comments and added still remaining studies.